# Short Culture of Bovine Hepatocytes Biopsied from Dairy Cows as a Model for Toxicological Studies—CYP 1A1 Activity Response to Zearalenone Treatment

**DOI:** 10.3390/ijms241512344

**Published:** 2023-08-02

**Authors:** Justyna Barć, Jadwiga Flaga, Anna Kozubek, Zygmunt Maciej Kowalski

**Affiliations:** 1Department of Animal Nutrition and Biotechnology, and Fisheries, University of Agriculture in Krakow, Al. Mickiewicza 24/28, 30-059 Krakow, Poland; jadwiga.flaga@urk.edu.pl (J.F.); rzkowals@cyf-kr.edu.pl (Z.M.K.); 2Department of Animal Physiology and Endocrinology, University of Agriculture in Krakow, Al. Mickiewicza 24/28, 30-059 Krakow, Poland; anna.kozubek@urk.edu.pl

**Keywords:** dairy cow, ketosis, methods, model, mycotoxins

## Abstract

This study presents a simple and cost-effective method for isolating hepatocytes from liver biopsies obtained from healthy and ketotic dairy cows, which can be utilized for studying cellular metabolism, drug toxicity, and hepatocyte-specific gene function and regulation. The expression of hepatocyte marker genes (*G6PC*, *ALB*, *CYP1A2*) was measured and found to be highest at 6 h post-isolation, with a subsequent decrease over time. Cells isolated from ketotic livers exhibited lower expression levels than those from healthy livers. Furthermore, for the functional characterization of ketotic hepatocytes, the cells were exposed to varying doses of zearalenone (ZEA). While doses of 10–50 µM did not affect cell viability, the highest dose of ZEA (100 µM) significantly decreased cell viability, as measured using XTT assay. Additionally, the potential induction of cytochrome P450 A1 (CYP1A1) by ZEA was found. Despite limitations such as a short-term culture, this model provides a useful tool for conducting toxicological research.

## 1. Introduction

The liver is the crucial organ responsible for countless actions, including detoxification and metabolic functions. Culture of primary hepatocytes has been used in studies concerning metabolic activities [1], properties of drugs [2], and nutrigenomics [3]. However, due to a lack of suitable cell lines for the liver of farm animals, research is limited to expensive and complex in vivo studies. Hence, establishing a good model of the primary culture of hepatocytes will allow conducting toxicological studies in different metabolic states of the liver including ketosis.

Ketosis, clinical (CK) or subclinical (SCK), is a major metabolic disorder in dairy cows resulting in significant economic losses [4]. Ketosis typically occurs in dairy cows in early lactation and is related to a state of negative energy balance (NEB) during the transition period [5,6]. Ketosis is closely associated with liver lipidosis, however, its pathogenesis, molecular basis, and ketotic liver’s potential response to toxins are still not fully understood. Therefore, establishing a functional model for research related to this disorder is needed and justified.

Among toxins to which the liver of dairy cattle may be exposed, mycotoxins appear to be the most common. They are a group of highly toxic compounds produced by fungi or yeasts, which commonly develop in places with low water availability, unsuitable for bacterial growth [7]. When the animal diet contains corn silage, one of the main mycotoxin contaminants is zearalenone (ZEA). However, cows may also be exposed to other mycotoxins and anti-nutrient factors [8]. As the liver of the ketotic cow’s primary objective is to cope with negative symptoms of energy imbalance, we assume that in the presence of impaired metabolic status causing excessive ketogenesis their ability to metabolize toxins may be impaired.

The aim of this study was to establish a simple technique to isolate and culture hepatocytes from biopsies obtained from the livers of dairy cows for a short-term culture suitable for toxicological experiments and to validate the technique for the study of the toxic properties of ZEA on the liver of healthy and ketotic cows.

## 2. Results and Discussion

The cells were isolated according to the combination of methods of Panda et al. [9] and Spotorno et al. [10] with an additional cold collagenase incubation stage to maximize cell yield from limited starting material. The viability of isolated hepatocytes differed between groups (*p* < 0.05) and was 92 ± 4.5%, 90.1 ± 3.9%, and 82.3 ± 3.5% for H, SCK, and CK groups, respectively. After 24 h of culture on collagen-coated plates, cells formed monolayer with clusters. Since no purification was performed, the cell population was not homogenous, with visible fibroblasts on the plates. Nevertheless, we observed the presence of the lipid droplets in cells, in which triacylglycerol and cholesteryl esters are stored, in each culture. Morphology of isolated hepatocytes was evaluated using inverted phase contrast microscopy. After isolation, undamaged hepatocytes were visible as spherical in shape, bright, translucent cells with well-contrasted borders. After adhesion (approximately 24 h of incubation), morphology had changed to a polygonal cell shape characteristic for hepatocytes. At this point, the medium was replaced, and we started to count the duration of culture on which further tests were performed.

The focus of research on the toxicological properties of drugs and their biotransformations in large ruminants has centered around studying cattle hepatocytes [11]. Several techniques have been employed to isolate these hepatocytes, with the Howard’s mechanical/enzymatic technique modified by Berry and Friend [12] being the most commonly used. This technique has been further refined by Seglen [13] through the development of a two-step collagenase perfusion method, which has also been modified by other researchers to achieve high yields of isolated hepatocytes [14]. However, these techniques require high concentrations of collagenase, extensive handling with perfusion and sieving apparatus, and are not suitable for use with small biopsy sections of the liver. Moreover, the methods mentioned above are specifically designed for collecting samples from substantial liver sections most often following animal slaughter or invasive procedure. The technique outlined in the article not only allows for the isolation of cells from in vivo samples but also simplifies the study of cells from animals experiencing different metabolic conditions. To our knowledge, our study is the first to show a method that allows for the isolation of hepatocytes from liver biopsies of ketotic cows. 

### 2.1. The mRNA Expression of Hepatocytes-Specific Genes

Expression of hepatocyte marker genes (*G6PC*, *ALB*, *CYP1 A2*) was evaluated after 6, 24, and 48 h of cell culture. The highest expression of all genes was noted after 6 h and then decreased with time. Relative expression of *G6PC* was statistically significantly higher in the hepatocytes isolated from H and SCK groups than in the hepatocytes from CK cows after 6 and 48 h of culture (*p* < 0.05); however, there were no difference between groups in *ALB* and *CYP1A2* mRNA expressions (Figure 1).

Examined genes were selected as markers to monitor cell functionality during culture. Expression of glucose-6-phosphatase (*G6PC*) plays a key role in glucose homeostasis. Studies suggest that in cows with ketosis, clinical or subclinical, there is an upregulation of *G6PC* expression and other genes involved in glucose metabolism [15]; however, our observation of a decrease in expression in cows with ketosis is probably due to the worse condition of cells obtained from ketotic cows. In hepatocyte culture, measuring albumin production is an important way to assess the function and viability of the cultured cells. The ability of hepatocytes to produce albumin is a key indicator of their differentiated phenotype and liver-specific function.

The gene *CYP1A2* codes for cytochrome P450, which plays a crucial role in the metabolism of drugs and xenobiotics. In the cultured hepatocytes, *CYP1A2* was also expressed, and its expression has been utilized as an indicator of the functional viability of the cells. To our knowledge, there are no studies that specifically investigate the relationship between *CYP1A2* and ketosis in cows. However, there are studies that examine the effect of ketosis on the expression of other cytochrome P450 genes in cows, for example, Peng et al. [16] found that the expression of *CYP2E1* was significantly upregulated in the liver of cows with clinical ketosis compared to healthy cows. Another study by Gaddis et al. [17] indicate a relationship between ketosis and *CYP26B1* gene expression. These findings suggest that the expression of cytochrome P450 genes may be altered in cows with ketosis, potentially affecting drug and xenobiotic metabolism.

The expression of genes specific to hepatocytes indicated that the cultured hepatocytes were functionally viable and that the simplified method used for their isolation was appropriate; however, their expression decreased quite quickly in culture, which means that cells isolated in this way are suitable only for short-term studies. Despite these limitations, we believe that the proposed model is sufficient to conduct research and may help to better understand the molecular basis of ketosis.

### 2.2. Functional Characterization of Ketotic Hepatocytes after ZEA Treatment

To determine the suitability of the model for toxicological studies, we performed XTT test to examine the cell viability after exposure to zearalenone at doses of 10, 20, 50, and 100 µM after 24 h of culture. ZEA did not reduce cell viability in the dose range 10–50 µM; however, the highest dose of ZEA (100 µM) significantly decreased the cell viability to 64.0%, 65.3%, and 63.6% in H, SCK, and CK groups, respectively (Figure 2). Taranu et al. [18] also found that zearalenone treatment led to a dose-dependent decrease in cell viability as measured using XTT assay on porcine epithelial cells (IPEC-1). The cytotoxic and oxidative effects of ZEA cells may be mediated through multiple signaling pathways, including ER stress [19] or generating ROS [20]. The lack of differences between the groups (H, SK, SCK) indicates a similar cell response regardless of the metabolic status of the liver.

Another test was performed to evaluate the model for toxicological studies that examined the possible induction of cytochrome P450 A1 (CYP 1A1) by ZEA in the dose range 10–100 µM after 24 h of culture. ZEA demonstrated a dose-dependent increase in the activity of CYP 1A1. In untreated cells of H, SCK, and CK groups, the enzyme activity was measured at 62.01 ± 6.63, 63.85 ± 5.25, and 61.52 ± 9.15 pmol per 100 μg protein min^−1^, respectively, for doses 20, 50, and 100 µM. However, with the highest ZEA dose (100 µM), the enzyme activity significantly rose to 195.52 ± 14.37, 187.37 ± 12.32, and 154.66 ± 11.83 pmol per 100 μg protein min^−1^ for H, SCK, and CK groups, respectively (Figure 3).

The study of Ayed-Boussema et al. [21] investigated the effect of zearalenone and its metabolites on the expression of cytochrome P450 enzymes (CYPs) in primary cultures of human hepatocytes. The authors found that zearalenone and its metabolites could induce the expression of CYP1A1 as well as other CYPs isoforms, similar to our study, where we demonstrated the induction of this enzyme in the hepatocytes of both healthy and ketotic cows.

Such a wide range of zearalenone doses was chosen for the study because ruminants are considered to be less susceptible to the harmful effects of mycotoxins in comparison to other livestock due to the presence of protozoal microbiota and rumen bacteria in the rumen fluid, which serve as the first line of defense against mycotoxins [22]; however, the protective barrier of the rumen can be disrupted by various factors such as changes in diet, animal diseases, or high levels of mycotoxin contamination. For instance, a decrease in ruminal pH, which is commonly caused by conditions such as rumen acidosis, can reduce the number of protozoa and compromise the antifungal capacity [23]. In addition, metabolic disorders such as ketosis can also impact ruminants’ resistance to mycotoxins as affected cows tend to eat very little, leading to significant alterations in the microflora of the rumen. Therefore, ruminant liver may actually be exposed to much higher doses of zearalenone than is commonly believed.

## 3. Materials and Methods

### 3.1. Animals

There were three groups of three early lactation dairy cows (10 days in milk average): healthy (H) with ketone bodies content in the blood < 1.4 mmol/L, subclinical ketosis (SCK) with ketone bodies between 1.4 and 3.0 mmol/L, and clinical ketosis (CK) with ketone bodies > 3.0 mmol/L. Assessment of ketone bodies in the blood (sampled from the tail vein, at 4–6 h after morning feed delivery) was carried out using a glucometer (Optimum Xido, Abbott Diabetes Care, Abbott Park, IL, USA) with the function of measuring ketone bodies level [24].

### 3.2. Liver Biopsies

Liver biopsies (0.5 to 1.5 g) for hepatocyte isolations were obtained by the vet surgeon using the procedure of Van den Top et al. [25]. Re-usable biopsy instruments (a cannula with a diameter of 0.9 cm, with trocar-tipped 52 cm length and a 0.8 cm diameter stainless steel stylet) were used. Biopsies were performed through the 11th intercostal space on the line between the elbow and hook. A skin area of 5 × 5 cm was clipped and disinfected with ethanol (70%) and a 3% solution of iodine. For local anesthesia, a 2% solution of Polocainum hydrochloricum cum adrenalino (10 mL per cow; Biowet, Puławy, Poland) was injected; then, a stab incision was made, and the biopsy needle was inserted in the direction of the opposite elbow to penetrate the liver. All biopsies were obtained under ultrasound guidance from the caudate lobe.

### 3.3. Isolation of Hepatocytes

Hepatocytes were collected and cultured according to the combination of two methods for isolation and purification of the hepatocytes, i.e., the simplified manual perfusion method described by Panda et al. [9] followed by the non-perifusion technique for cell isolation described by Spotorno et al. [10] with slight modifications. Tissue obtained by biopsy was aseptically transferred to a flask containing Dulbecco’s phosphate buffered saline (DPBS, Gibco, Billings, MT, USA) with antibiotic–antimycotic solution (Sigma Aldrich, St. Louis, MO, USA) in a sterile area and transported to the laboratory on ice in less than 60 min. In sterile conditions, liver samples were manually perfused with 100 mL of pre-cooled (4 °C) Ca^2+^ and Mg^2+^ free 33 mM HEPES [4-(2-hydroxylethyl)-1-piperazineethanesulfonic acid] buffer (pH = 7.6, Gibco), containing 0.5 mM EGTA (ethylene glycol-bis(β-aminoethyl ether)-N,N,N′,N′-tetraacetic acid; Sigma Aldrich), using a 50 mL sterile syringe to remove the blood clots. Then, the tissue was minced with scissors, homogenized, and washed several times with DPBS (without EGTA). After a last wash with DPBS, the samples were transferred to a conical flask with a cold collagenase solution (Collagenase II, 100 U/mL in HBSS-Hepes buffer), and softly stirred for 12 min. At the end of the incubation, FBS (Cytogen, Seoul, Republic of Korea) with ice cold HBSS (Gibco) was added. The cell suspension was then filtrated through cheesecloth and centrifuged at 150× *g* for 5 min at 4 °C with addition of DNase solution. The supernatant was discarded, and the cell pellet was washed with DPBS and centrifuged twice at 150× *g* for 5 min each. Finally, the pellet was suspended in DPBS.

### 3.4. Cell Culture

Cells were seeded on dry collagen-coated plates at a density of 4.5–1 × 10^5^ of viable cells/cm^2^, in growth medium with 10% of FBS. The cultures were maintained at 38.5 °C, in a humidified atmosphere, containing 5% CO_2_ for 24 h to allow attachment. After 24 h, the medium was replaced with fresh medium, and from this point, the start of the culture was counted and then stopped after 6, 24, and 48 h intervals. To test the model for toxicological studies, on the basis of CYP1A1 activity, the cells after attachment were cultured with or without ZEA (Sigma Aldrich) in doses of 10, 20, 50, and 100 μM for 24 h.

### 3.5. qPCR

RNA isolation and cDNA synthesis were performed using the TaqMan Gene Expression Cells-to-CT kit (Applied Biosystems, Foster City, CA, USA) according to the manufacturer’s protocol. The resulting pre-amplified cDNA preparations were analyzed by real-time PCR in a StepOnePlus Real-time PCR System (Applied Biosystems, Foster City, CA, USA) using TaqMan Gene Expression Assays and TaqMan Gene Expression Master Mix containing ROX (Applied Biosystems, Foster City, CA, USA) for the following genes: albumin (*ALB*, Bt03231287_m1), glucose-6-phosphatase (*G6PS*, Bt03242914_m1), and cytochrome P450 1A2 (*CYP1A2*, Bt03817607_m1). The following PCR conditions were used: incubation for 2 min at 50 °C followed by incubation for 10 min at 95 °C and 40 cycles (denaturation step: 15 s at 95 °C; annealing/elongation step: 60 s at 60 °C). Obtained results were normalized to *GAPDH* (Bt03210912_g1) as a reference gene.

### 3.6. Oil Red Staining

Medium was removed from hepatocytes culture and then 10% formalin was added and incubated for 5 min at room temperature. After that, the formalin was discarded and replaced with the fresh one for a further incubation for 1 h. Then, formalin was removed and the cells were washed with 60% isopropanol and allowed to dry. Working oil red stain (mixing of stock oil red (Sigma Aldrich) and water in 3:2 ratio) was then added and incubated for 10 min. Then, oil red stain was removed, and the culture was washed with distilled water four times and stained lipid droplets were observed under inverted light microscope (Leica LED, DM IL, Wetzlar, Germany).

### 3.7. XTT Cell Viability Assay

The XTT sodium salt [2,3-bis(2-methoxy-4-nitro-5-sulfophenyl)-2H-tetrazolium-5-carb xanilide inner salt] assay was used to measure the viability of cells. In the test, tetrazolium salts are reduced to formazan by mitochondrial succinate dehydrogenase, an enzyme that is only active in cells with an intact metabolism and respiratory chain. The formazan can be quantified photometrically and correlates with the metabolic activity and number of viable cells. The XTT assay (Xenometrix, Allschwil, Switzerland) was performed according to the manufacturer’s instructions. The XTT reduction in the cultures was determined at the beginning of experiment after cell attachment (0 h) and at the end of incubation with ZEA after 24 h, by measuring the absorbance at 450 nm, using an absorbance ELISA microplate reader (Infinite M200, Magellan Tecan, Männedorf, Switzerland). All samples were run in quadruplicate in the same assay. Results were analyzed using Magellan™–Data Analysis Software version 7.2.

### 3.8. CYP1A1 Activity

To determine CYP1A1 activity, BSA solution (1.33 mg/mL in 50 mM Tris, pH 7.2; final reaction concentration, Sigma Aldrich) and ethoxyresorufin solution (10 μM in 50 mM Tris, pH 7.2; final reaction concentration, Sigma Aldrich) was added to each well. Blank wells received BSA solution, ethoxyresorufin solution, and Tris buffer (50 mM, pH 7.2). Plates were incubated for 15 min at 37 °C with gentle shaking. To start the reaction, NADPH solution (1.67 mM in 50 mM Tris, pH 7.2; final reaction concentration, Sigma Aldrich) was added to each well, excluding the blank wells, and the plates were incubated at room temperature without shaking. Fluorescence resulted from the CYP1A-mediated oxidation of the non-fluorescent ethoxyresorufin to the fluorescent resorufin was read at 15 min intervals for up to 2 h with a fluorescence plate reader (Infinite M200, Magellan Tecan, Männedorf, Switzerland) at a 530 nm excitation wavelength and a 590 nm emission wavelength. After 2 h, 90 μg of fluorescamine (in acetonitrile, Sigma Aldrich) was added to each well to stop the reaction, and the protein concentration of each well was determined with a 400 nm excitation filter and 460 nm emission filter.

### 3.9. Statistical Analysis

Each result is presented as the mean ± SD of 12 results (three animals from each of the three groups, and four independent cultures performed from each biopsy). The results were subjected to the one-way or two-way analysis of variance (ANOVA). The mRNA expression of hepatocyte-specific genes was subjected to the one-way ANOVA, where the health status (H, SCK, K) was a fixed effect (GraphPad Prism version 5.0, GraphPad Software Inc., San Diego, USA). The data on functional characterization of hepatocytes were subjected to the two-way ANOVA, with the health status and the ZEA dose as the fixed effects, followed by Tukey’s HSD (honestly significant difference) multiple-range test. Data for individual times (6 h, 24 h, 48 h) were analyzed separately. Significant differences (at *p* < 0.05) were indicated with different letters.

## 4. Conclusions

In this study, we have developed a simple and cost-effective method to isolate hepatocytes from small liver biopsies obtained from both healthy and ketotic cows. The primary cultures of hepatocytes obtained can be used for studying cellular metabolism, drug toxicity, and hepatocyte-specific gene function and regulation. We have observed that the expression of hepatocyte marker genes (*G6PC*, *ALB*, *CYP1A2*) was highest at 6 h post-isolation, but decreased over time. We also found that cells obtained from ketotic cows expressed these genes at lower levels than those from healthy cows, likely due to their poorer metabolic conditions. Furthermore, we evaluated the functional characterization of ketotic hepatocytes by exposing them to varying doses of zearalenone (ZEA). Our results showed that high doses of ZEA (100 µM) reduced cell viability, while lower doses (10–50 µM) did not affect cell viability, as measured using XTT assay. Additionally, we investigated the potential induction of cytochrome P450 A1 (CYP1A1) by ZEA and found that it does indeed induce this protein. The innovative aspect of this technique lies in its minimally invasive approach to liver biopsy, eliminating the need to euthanize the animal. This allows studying animals in various metabolic states, as demonstrated in the case of ketosis. Consequently, it opens up new avenues to explore cellular changes in the liver influenced by different metabolic conditions. Despite the limitations of our study, such as the short-term culture period, we believe that this model can help to enhance our understanding of the molecular basis of ketosis and provide a valuable tool for conducting research in the field of toxicology and nutrigenomics.

## Figures and Tables

**Figure 1 ijms-24-12344-f001:**
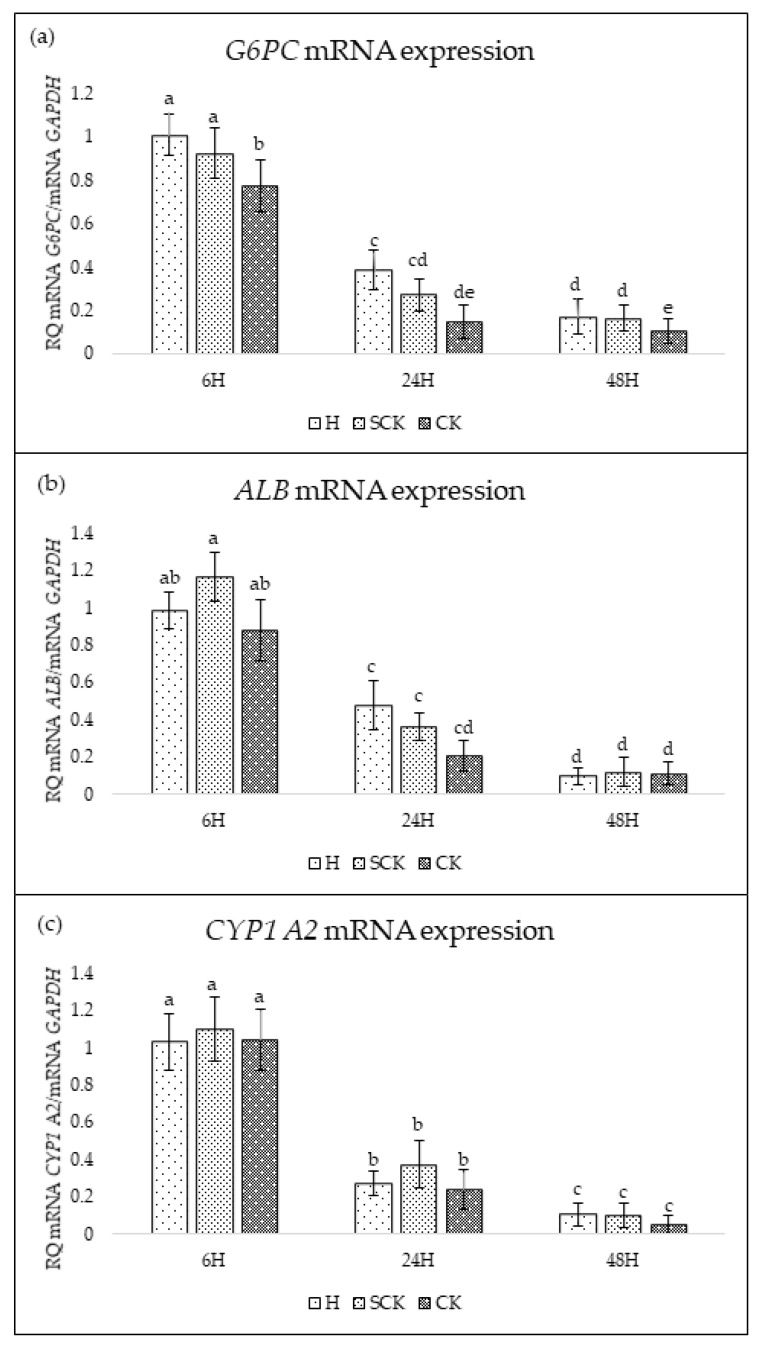
Relative expression of *G6PC* (**a**), *ALB* (**b**), and *CYP1 A2* (**c**). mRNA expression was determined using qPCR and expressed as mean ± SD. All data was normalized to H group after 6 h as a reference in 2(−delta delta C(T)) calculation. Statistically significant differences between groups are indicated with different letters; the same letters indicating no significant differences.

**Figure 2 ijms-24-12344-f002:**
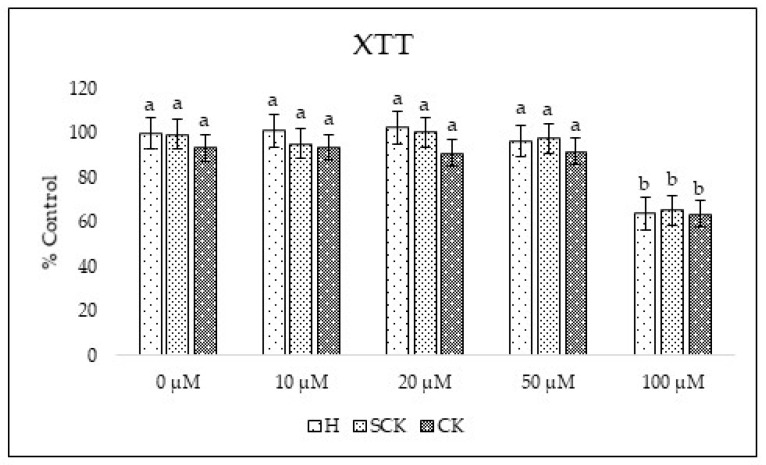
Effect of ZEA on cell viability measured using XTT test after 24 h of culture showed as percentage of control (untreated cells from H group). Statistically significant differences between groups are indicated with different letters; the same letters indicating no significant differences.

**Figure 3 ijms-24-12344-f003:**
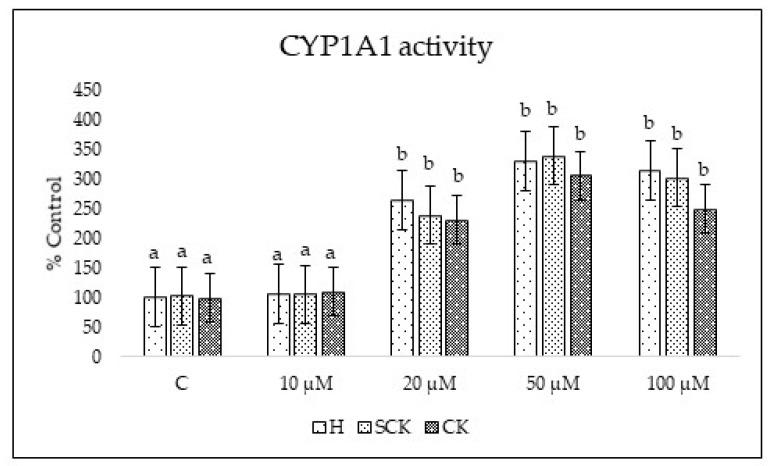
Dose-dependent action of ZEA (10, 20, 50, and 100 µM) on CYP1 A1 activity after 24 h of culture showed as percentage of control (untreated cells from H group). Statistically significant differences between groups are indicated with different letters; the same letters indicating no significant differences.

## Data Availability

Data available upon request.

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
