# Peer review of "Short Culture of Bovine Hepatocytes Biopsied from Dairy Cows as a Model for Toxicological Studies—CYP 1A1 Activity Response to Zearalenone Treatment"

_ijms, 2023, doi:10.3390/ijms241512344_

Round 1
Reviewer 1 Report
The manuscript entitled "Short culture of bovine hepatocytes biopsed from dairy cows as a model for toxicological studies – CYP 1A1 activity response for zearalenone treatment" offers an interesting tool for hepatic studies. I would like to show some minor questions which should be ammended
There are some abbreviations along the text that are not explained, for instance DIM (line 160) or DMIL (line 225). Please check the text.
There is a typo ((GraphPad Prism 5). .) in line 259
Please, rewrite the sentence in line 130-134
Please give a reference for sentence "The cytotoxic and oxidative effects of ZEA cells may be mediated through multiple signaling pathways, including the ERK and AKT pathways" or discuss it further in line 120
I think English is fine
Author Response
Thank you for all your comments and suggestions. You can find my responses below.
There are some abbreviations along the text that are not explained, for instance DIM (line 160) or DMIL (line 225). Please check the text.
DIM was replaced by “days in milk” (line 169)
DM IL is a model of microscope used in research, its full name is Leica DM IL Inverted Fluorescence Phase Contrast Microscope, I assume that is producers letter designation, not an abbreviation.
There is a typo ((GraphPad Prism 5). .) in line 259
A typo has been corrected
Please, rewrite the sentence in line 130-134
The sentence has been rewritten to be clearer (line 139-143)
Please give a reference for sentence "The cytotoxic and oxidative effects of ZEA cells may be mediated through multiple signaling pathways, including the ERK and AKT pathways" or discuss it further in line 120
The sentence has been changed to make it clearer and the relevant references have been added in the text (line 127-129)
Reviewer 2 Report
This manuscript proposes a new method for isolating hepatocytes from cows. Overall the manuscript is well written and does contribute to the field. I believe the manuscript could be improved with more discussion of previously used isolation techniques. The authors briefly mention that previous techniques "are not suitable for use with small biopsy sections". A more expanded discussion providing the advantages and disadvantages of this technique compared to previously used techniques could increase the impact of this paper.
The authors also indicate that this technique was derived from previous methods by Panda et al. and Spotorno et al. I believe a further discussion of the new technique versus these previous techniques could help to indicate the novelty of the new proposed technique.
A couple other minor notes throughout the manuscript:
Line 79-80 indicates higher expression of ALB in H and SCK groups. However, according to the figure, under each of the time intervals the ALB expression is not significantly higher for the H and SCK groups (i.e., all expressions after 6H are marked with an a, indicating no significant difference).
The figures appear somewhat faded due to the lighter font and lines used. I would suggest a darker font and lines to make the figures stand out more.
Overall quality was good, but I did notice a few minor things.
Line 78-79: should be "and then decreased with time"
Line 81: elsewhere this group is referred to as "CK". I believe this should be kept consistent throughout the paper.
Line 194-195: Usually centrifugal force is indicated as "x g" to distinguish from gram units.
Line 203-204 is slightly unclear when referring to the timing of the ZEA treatment. This would be 24 hrs from the start of the culture, not 24 hrs after the 48 hr interval, correct?
Author Response
Thank you for all your comments and suggestions. You can find my responses below.
This manuscript proposes a new method for isolating hepatocytes from cows. Overall the manuscript is well written and does contribute to the field. I believe the manuscript could be improved with more discussion of previously used isolation techniques. The authors briefly mention that previous techniques "are not suitable for use with small biopsy sections". A more expanded discussion providing the advantages and disadvantages of this technique compared to previously used techniques could increase the impact of this paper.
The authors also indicate that this technique was derived from previous methods by Panda et al. and Spotorno et al. I believe a further discussion of the new technique versus these previous techniques could help to indicate the novelty of the new proposed technique.
Thank you for these comments. As suggested, descriptive sentences have been added to make more clear why our technique differs from those known so far, which is, first of all, a small amount of material is enough to obtain enough cells to conduct research and, additionally, the possibility of collecting material from animals in various metabolic states without the need to euthanize them (lines 53-55, 75-80 and 286-290)
A couple other minor notes throughout the manuscript:
Line 79-80 indicates higher expression of ALB in H and SCK groups. However, according to the figure, under each of the time intervals the ALB expression is not significantly higher for the H and SCK groups (i.e., all expressions after 6H are marked with an a, indicating no significant difference).
Thank you, it was obviously a mistake. It has been corrected.
The figures appear somewhat faded due to the lighter font and lines used. I would suggest a darker font and lines to make the figures stand out more.
Figures have been changed according to suggestions
Overall quality was good, but I did notice a few minor things.
Line 78-79: should be "and then decreased with time"
It has been corrected.
Line 81: elsewhere this group is referred to as "CK". I believe this should be kept consistent throughout the paper.
It has been corrected.
Line 194-195: Usually centrifugal force is indicated as "x g" to distinguish from gram units.
It has been corrected.
Line 203-204 is slightly unclear when referring to the timing of the ZEA treatment. This would be 24 hrs from the start of the culture, not 24 hrs after the 48 hr interval, correct?
Separate cell cultures were conducted to test the effect of zearalenone. 24 hours refers to the incubation time with zearalenone, which was added after cells attachment. This information has been added in the text (line 211-213)